# Eco-Friendly Approach for Graphene Oxide Synthesis by Modified Hummers Method

**DOI:** 10.3390/ma15207228

**Published:** 2022-10-17

**Authors:** Néstor Méndez-Lozano, Francisco Pérez-Reynoso, Carlos González-Gutiérrez

**Affiliations:** Universidad del Valle de México, Campus Queretaro, Blvd. Juriquilla No. 1000 A Del. Santa Rosa Jáuregui, Queretaro C.P. 76230, Mexico

**Keywords:** Hummers method, eco-friendly synthesis, graphene oxide, nanostructure sheet

## Abstract

The aim of this study is to produce graphene oxide using a modified Hummers method without using sodium nitrate. This modification eliminates the production of toxic gases. Two drying temperatures, 60 °C and 90 °C, were used. Material was characterized by X-Ray Diffraction, Fourier Transform Infrared Spectroscopy, Raman Spectroscopy and Scanning Electron Microscopy. FTIR study shows various functional groups such as hydroxyl, carboxyl and carbonyl. The XRD results show that the space between the layers of GO60 is slightly larger than that for GO90. SEM images show a homogeneous network of graphene oxide layers of ≈6 to ≈9 nm. The procedure described has an environmentally friendly approach.

## 1. Introduction

Graphene has excellent mechanical, electronic, optical and thermal properties. It has a unique two-dimensional structure one atom thick [1]. Many researchers have been interested in investigating this two-dimensional (2D) form of carbon because it has become a relevant topic for the development of materials with many applications [2]. As reported in the literature, graphene has a large specific surface area [3], an efficient electron mobility (200,000 cm^2^ v^−^^1^ s^−^^1^) [4,5], a high Young’s modulus (1 TPa) [6], and good thermal conductivity (4.84 × 10^3^ to 5.30 × 10^3^ W/mK) [7]. Graphene oxide (GO) can be manufactured or self-assembled into materials with controlled compositions and microstructures for different applications [8]. Previous work has reported the use of graphene oxide combined with fullerene in thin-film form to produce lightweight three-dimensional hybrid structures with high surface area [9]. The arrangement of other molecules within graphene oxide layers has shown that multilayer structures exhibit high biocatalytic activity [10]. The Langmuir–Blodgett process has recently been used for the production of graphene oxide by which a uniform dispersion and controllable development of graphene oxide flakes has been achieved [11].

The most important and widely applied method for GO synthesis is that developed by Hummers and Offeman [12]. This method has three important advantages over other techniques. First, the reaction is complete in a few hours, second, potassium chlorate can be replaced by potassium permanganate for a safer reaction, and third, the use of sodium nitrate eliminates acid mist formation. However, the method also has some defects, since in the oxidation process, some toxic gases such as nitrogen dioxide and dinitrogen tetroxide are released. In addition, sodium and nitrate ions are difficult to remove from the wastewater formed during the process of synthesis and purification of graphene oxide.

In previous works, the Hummers method has been improved by excluding sodium nitrate and increasing the amount of potassium permanganate, carrying out the reaction in a single mixture [13]. With this modification, it is possible to increase the performance of the reaction and reduce the release of toxic gases; also, phosphoric acid is introduced in the reaction system. Previous research has reported that the mixture of sulfuric acid and nitric acid used in the Hummers method acts as a “chemical scissors” for graphene planes that facilitates the penetration of the oxidation solution [14].

On the other hand, potassium permanganate can achieve the complete intercalation of graphite, forming graphite bisulfate [15,16]. This interaction ensures the effective penetration of potassium permanganate into the graphene layers for graphite oxidation. Due to this, potassium permanganate replaces the function of sodium nitrate, so it is not necessary for the reaction. In this investigation, we show an easy synthesis route to produce GO using a low-cost and environmentally friendly modified Hummers method. In addition, the synthesis route is highly reproducible in obtaining graphene oxide for its subsequent reduction (rOG) for possible biocatalytic applications as reported in previous works.

## 2. Materials and Methods

### Synthesis of GO

Materials included: natural graphite (99%) supplied by Aldrich chemistry, potassium permanganate (KMnO_4_) supplied by Sigma Aldrich, sulfuric acid (H_2_SO_4_) supplied by Jalmex, hydrochloric acid (HCl) supplied by Sigma Aldrich, and hydrogen peroxide (H_2_O_2_) supplied by J.T Baker. All the reagents were obtained in the city of Queretaro, Mexico.

The synthesis process for obtaining graphene oxide is described below. Two glycerin baths are preheated to 45 °C and 98 °C, respectively. Then, 1 g of graphite was added in a ball flask in a cold bath for 5 min with 23 mL of sulfuric acid (H_2_SO_4_), it was stirred for 5 min. Subsequently, potassium permanganate (KMnO_4_) was added and placed in the glycerin bath at 45 °C for 2 h. After 2 h, the mixture was transferred to a glycerin bath at 98 °C, adding 46 mL of distilled water at room temperature; then, it was kept for 15 min. After 15 min, 140 mL of hot water was added along with 10 mL of hydrogen peroxide (H_2_O_2_). The mixture obtained was emptied into the strainer to filter by vacuum. The sample was removed and placed in 6 jars with 1 g of sample. Finally, 2 mL of hydrochloric acid (HCl) and distilled water were added to wash the samples by centrifugation. The final sample was placed in a petri dish to be dried in oven at 60 °C and 90 °C for 24 h. In Table 1, the number of washes, the centrifugation revolutions and the time for each sample is shown. In addition, in Figure 1, the results after washing and after drying, respectively, are observed. Finally, in Figure 2, the synthesis process is shown in a flow chart. Table 2 shows a comparison between the traditional synthesis method and the modified Hummers method used in this work.

## 3. Results and Discussion

### 3.1. X-ray Diffraction (XRD)

In this study, X-Ray Diffraction (XRD) was used to determine the crystal structure and verify the spacing between the GO layers.

The XRD pattern for the sample dried at 60 °C, GO60-1 and GO60-2 is presented in Figure 3. This sample exhibits a diffraction peak at 9.28° due to the (002) plane of GO [17]. In addition, a small peak at ≈26° is observed; according to the literature, this peak corresponds to graphite. When the graphite is oxidized, the diffraction peak should change from ≈26° to ≈11°; this agrees with the results observed in Figure 3.

On the other hand, the XRD pattern for the sample dried at 90 °C is presented in Figure 4 GO90-1 and GO90-2. This sample presents a diffraction peak at 9.6°, which is slightly different from the GO60 sample. In GO90-1, a small peak at ≈26 corresponding to graphite can also be observed; however, in GO90-2, this peak is no longer present, which suggests that in said sample, all the graphite was oxidized to become GO. In addition, the intensity of it is three times greater compared to GO90-1, which suggests a greater number of planes (002) in that direction. This difference could be due to the increase in the drying temperature compared to the GO60 sample.

In both samples, the observed peaks are sharp, which indicates that the graphite was completely oxidized by this method. The spacing between the GO layers was calculated using Bragg’s law [18].
λ=2dsinθ
where n is the diffraction series and λ is the X-ray wavelength $\left(0.154\;\mathrm{nm}\right)$. The spacing between the GO60 and GO90 layers was 0.95 nm and 0.92 nm, respectively, according to Bragg’s law. Both spacings are slightly different, since a different drying temperature was used. Usually, the interlayer spacing d of graphene oxide is in the range of 0.6–1.0 nm and is controlled according to the degree of oxidation of graphite and the number of intercalated water molecules in the interlayer space [19]. In previous works, it has been reported that the increase in the space between the layers is due to the intercalated functional group of oxygen and water molecule in the structure of the carbon layer [20].

On the other hand, Zeng et al. mention that the increase is related to the weaker Van der Waals bond formed by the epoxyl, hydroxyl, carbonyl and carboxyl groups in the basal planes [21].

### 3.2. Fourier Transform Infrared Spectroscopy (FTIR)

The FTIR spectra of the GO60 and GO90 samples are shown in Figure 5. The spectra consist of vibrational groups of GO that include carbonyl (C=O), aromatic (C=C), and hydroxyl (O-H) groups; these groups appear in Table 3. The O-H stretching vibrations in the region of 3500–3000 cm^−^^1^ are attributed to the carboxyl and hydroxyl groups of the residual water present in the GO samples. These hydrophilic functional groups containing oxygen provide GO samples with good dispersibility in water [22].

The peak in 1730 cm^−^^1^ is due to the ketone group (C=O), while the peak at 1567 cm^−^^1^ is the main graphitic domain and is due to sp2 hybridization [23]. Finally, the band at 1100 indicates the C-O stretching of the epoxy groups [24]. These results suggest that graphite powder is successfully oxidized in the presence of acid with potassium permanganate (KMnO_4_).

### 3.3. Raman Analysis

The Raman spectrum of GO60 and G090 is shown in Figure 6. Both Raman spectra contain bands marked as D and G bands. Peak D appears at ≈1300 cm^−^^1^, while peak G appears at ≈1600 cm^−^^1^. The G band is associated with graphitic carbons, and the D band is related to structural defects or partially disordered graphitic domains [25].

In both spectra, the D bands are strong, which confirms the distortions of the graphene basal plane lattice. In addition, the G band is prominent for sp2 carbon lattices. Ferrari et al. mention that the D band reveals disorders of crystalline materials and defects associated with vacancies and grains [26]. On the other hand, the G peak corresponds to the optical phonons in the center of the Brillouin zone that result from the stretching of the bond of the sp2 carbon pairs in the rings as well as in the chains [27]. Therefore, the intensity of the ratio of I_D_/I_G_ was calculated for both samples, resulting in 1.26 and 1.2 for GO60 and GO90, respectively. These results provide evidence of the degree of functionalization of graphene oxide [28]. According to the literature, it is possible to obtain the number of layers in the graphene oxide flakes from the position and shape of the D band in the Raman spectra [29]. Our results show a few layers in the flakes in the order of 3. Silva et al. mention that the D band gives us information about the exfoliation of graphene, while the G band provides information about the number of layers [30].

### 3.4. Scanning Electron Microscopy (SEM)

Figure 7 shows SEM images of sample GO60 with magnifications of (a) ×15,000, (b) ×25,000 and (c) ×100,000. On the other hand, in Figure 8, SEM images of the GO90 sample are shown at the same magnifications of the GO60 sample.

Aggregated leaves are observed in both micrographs; a structure of ultrafine and homogeneous layers can be observed. Sheets are folded or continuous, and it is possible to distinguish the edges of individual sheets, including crooked and wrinkled areas. The SEM images revealed that our material consists of thin and wrinkled sheets; furthermore, they are randomly aggregated and closely associated with each other, forming a disordered solid. It was observed that the folded regions (Figure 7c) have an average width of ≈6 nm, while the fold thickness (Figure 8c) has an average of ≈9 nm. These measurements represent the thickness of the network of graphene oxide layers.

High-resolution SEM data suggest the presence of individual leaves in GO60 and GO90. The measured value for fold thickness in both samples suggests a confidence limit of approximately ±1 nm. The absence of charge during the SEM image indicates that the network of graphene oxide-based sheets and the individual sheets is electrical [31].

### 3.5. Energy-Dispersive Spectroscopy (EDS)

In order to obtain the elemental composition of the samples, EDS analyses were performed. A sweep over different regions was made, showing in both samples the presence of carbon and oxygen as predominant elements. Other impurities such as potassium, chlorine, sulfur and silicon are also present in less quantity. The presence of these elements is due to the precursors used in the synthesis process; however, the washings carried out on the samples reduce the presence of impurities. The presence of aluminum is due to the sample holder used in the study. Figure 9 shows the region analyzed and the spectrum obtained in each sample at an amplification of ×5000. Table 4 shows the elemental composition for GO60 and GO90.

## 4. Conclusions

In conclusion, we developed a modified Hummers method without using sodium nitrate (NaNO_3_) to obtain graphene oxide. With this method, we eliminate the generation of toxic gases and simplify the procedure, thus reducing the cost of GO synthesis. GO characterizations indicate that the products have similar chemical structure, thickness and dimensions. The exclusion of sodium nitrate produces the same characteristic of graphene oxide and does not affect the overall reaction yield. The modified Hummers method that we developed can be used to prepare GO on a large scale and is the first step to obtain pure graphene and all its derivatives. The synthesis described in this work has an environmentally friendly approach. As such, graphene oxide can find uses in a variety of applications such as energy storage and as a conductive filler material in composite materials.

## Figures and Tables

**Figure 1 materials-15-07228-f001:**
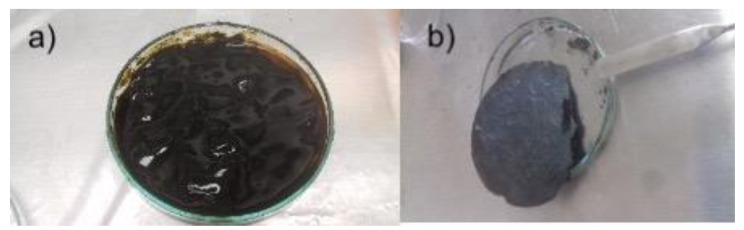
GO synthesis by the Hummers method: (**a**) after washing, (**b**) after drying.

**Figure 2 materials-15-07228-f002:**
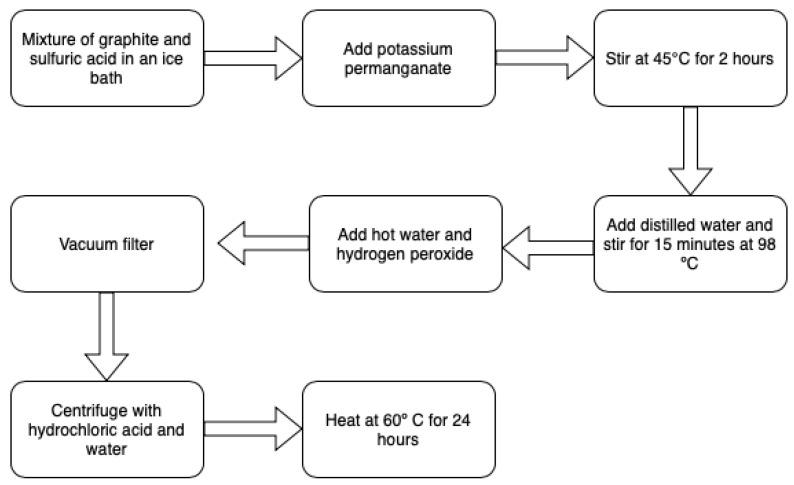
GO synthesis by the Hummers method at 60 °C and 90 °C for 24 h.

**Figure 3 materials-15-07228-f003:**
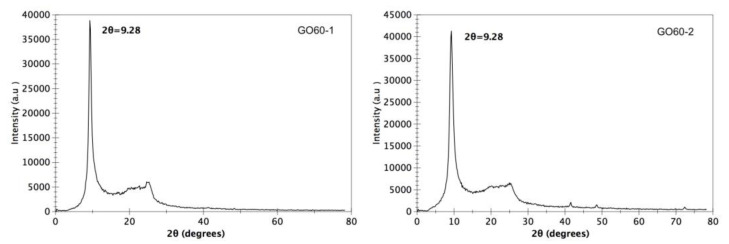
XRD pattern of graphene oxide dried at 60 °C.

**Figure 4 materials-15-07228-f004:**
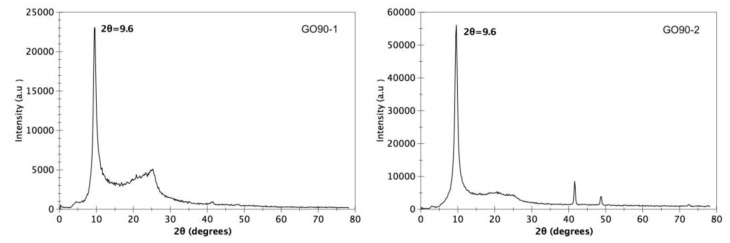
XRD pattern of graphene oxide dried at 90 °C.

**Figure 5 materials-15-07228-f005:**
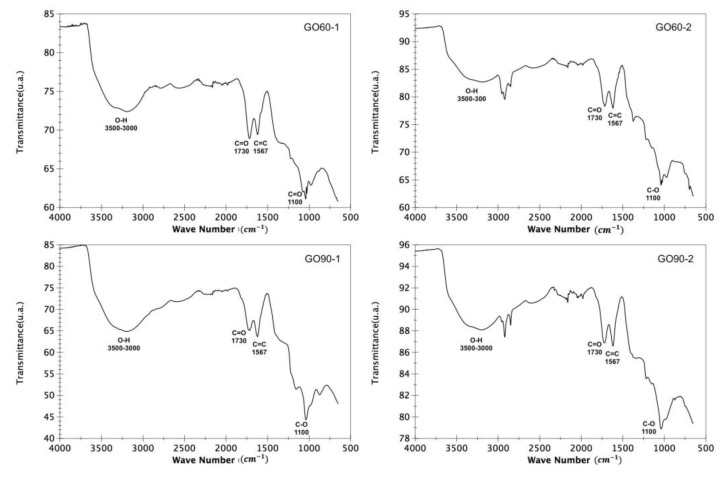
FTIR spectra of the GO60 and GO90 samples. Characteristic vibrational modes of graphene oxide are observed.

**Figure 6 materials-15-07228-f006:**
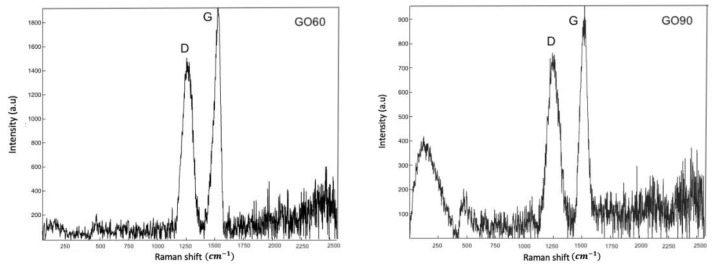
Raman spectra of graphene oxide GO60 and GO90.

**Figure 7 materials-15-07228-f007:**
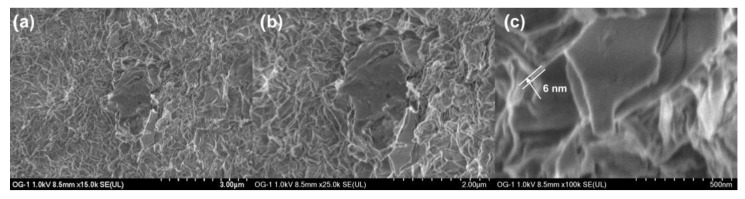
SEM image of GO60 sheets (**a**) ×15,000, (**b**) ×25,000 and (**c**) ×100,000. Network thickness of graphene oxide layers of ≈6 nm on average.

**Figure 8 materials-15-07228-f008:**
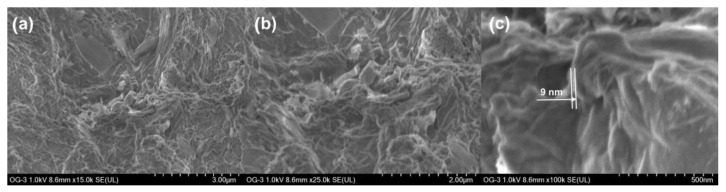
SEM image of GO90 sheets (**a**) ×15,000, (**b**) ×25,000 and (**c**) ×100,000. Network thickness of graphene oxide layers of ≈9 nm on average.

**Figure 9 materials-15-07228-f009:**
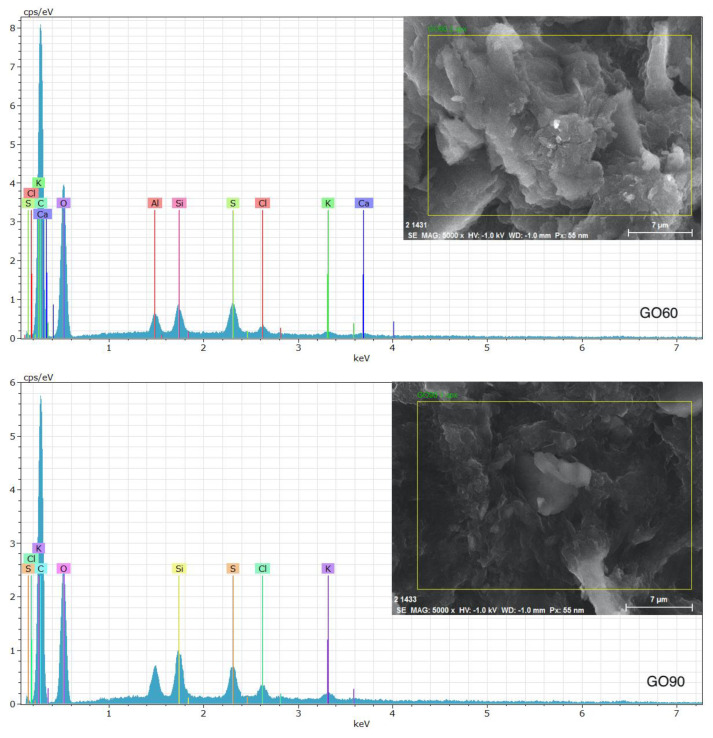
EDS analysis shows the elemental composition of samples GO60 and GO90.

**Table 1 materials-15-07228-t001:** Centrifugation washes.

Washed	Cycles (rpm)	Time (min)
1 to 10	300	5
11	300	10
12 to 14	300	5
15	300	10
16 to 19	300	5
20	300	10

**Table 2 materials-15-07228-t002:** Comparison of traditional synthesis vs. Hummers method for graphene oxide.

Method	Oxidants	Advantages	Disadvantages
Traditional	KCIO_3_ (NaCIO_3_) ,HNO_3_, H_2_SO_4_	-	Time-consuming and dangerous method. The risk of explosions are a constant danger.
Hummers modified	KMnO_4_, H_2_SO_4_ ,NaNO_3_	Improved level of oxidation and, therefore, product performance.	Separation and purification processes are tedious. Highly time- consuming.

**Table 3 materials-15-07228-t003:** Characteristic vibrational modes and their energies of GO60 and GO90.

Band Shift (cm^−1^)	Functional Group
3500–3000	O-H
1730	C=O
1567	C=C
1100	C-O

**Table 4 materials-15-07228-t004:** Elemental composition.

Sample	Element	wt %
GO60	C	49.29
	O	42.09
	Si	6.73
	S	1.13
	Cl	0.25
	K	0.46
GO90	C	51.14
	O	43.23
	Si	2.60
	S	1.42
	Cl	0.53
	K	0.77

## Data Availability

Not applicable.

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
