# Peer review of "Eco-Friendly Approach for Graphene Oxide Synthesis by Modified Hummers Method"

_materials, 2022, doi:10.3390/ma15207228_

Round 1
Reviewer 1 Report
The novelty of the paper is very low. Authors have only reported materials characterization of the synthesized materials by well know method known as the modified Hummer ́s method without using sodium nitrate.
Author Response
Editor-in-Chief
Materials
We sincerely appreciate that you have taken the time to review our work Eco-friendly approach for graphene oxide synthesis by modified Hummer’s method. We welcome your comments which are all valuable and helpful in improving the quality of our article.
The observations have been carefully studied, and, in the same way, the proper corrections have been made, hoping that they will meet your expectations and thus have your approval. The following pages of this document respond to the indications of each reviewer.
The text highlighted in green corresponds to each of the observations made by the reviewers, while the text highlighted in yellow shows the correction made by the authors. The authors welcome any further observations or future comments.
Dr. Néstor Méndez Lozano
(corresponding autor)
Reviewer 1
We sincerely appreciate your taking the time to review our work. We greatly appreciate each of your comments that help improve the quality of our work. Considering your comments, we have made the following responses:
The novelty of the paper is very low. Authors have only reported materials characterization of the synthesized materials by well know method known as the modified Hummer ́s method without using sodium nitrate.
The introduction was improved. Elemental composition (EDS) results were added, and the conclusions were improved

Reviewer 2 Report
The authors present an interesting and well written manuscript on the synthesis to produce GO using a low-cost and environmentally friendly modified hummers method. The rationale for conducting this work is well-justified and the paper aligns nicely with the scopes of the journal. The paper is well-organized and flows smoothly. I would recommend the following points before publication:
1. In the Raman analysis, please include the ID/IG ratio that shows the functionalization degree. Also from the Raman analysis you can calculate the number of flakes.
2. The thickness of the GO nanosheet is of significance importance. Please include an AFM measurement and statistical analysis of thickness and lateral dimension. This is very important due to the novelty of the synthesis; the readers must have all the information about the quality/morphology of the nanosheets. From the SEM you cannot have an estimation of the thickness. In your images you have a network not an isolated nanosheet.
3. You should write a paragraph in the introduction about the novelty of your work. It’s not very wise for your good work to have only 2-3 sentences.
4. The following references, they will help you enhancing the quality of your MS. Please include them.
· Graphene Oxide–Cytochrome c Multilayered Structures for Biocatalytic Applications: Decrypting the Role of Surfactant in Langmuir–Schaefer Layer Deposition (doi: 10.1021/acsami.2c03944)
· Controlled deposition of fullerene derivatives within a graphene template by means of a modified Langmuir-Schaefer method (doi: 10.1016/j.jcis.2018.04.049)
· Mapping of graphene oxide and single layer graphene flakes—defects annealing and healing (doi: 10.3389/fmats.2018.00037)
Author Response
Editor-in-Chief
Materials
We sincerely appreciate that you have taken the time to review our work Eco-friendly approach for graphene oxide synthesis by modified Hummer’s method. We welcome your comments which are all valuable and helpful in improving the quality of our article.
The observations have been carefully studied, and, in the same way, the proper corrections have been made, hoping that they will meet your expectations and thus have your approval. The following pages of this document respond to the indications of each reviewer.
The text highlighted in green corresponds to each of the observations made by the reviewers, while the text highlighted in yellow shows the correction made by the authors. The authors welcome any further observations or future comments.
Dr. Néstor Méndez Lozano
(corresponding autor)
Reviewer 2
We sincerely appreciate your taking the time to review our work. We greatly appreciate each of your comments that help improve the quality of our work. Considering your comments, we have made the following responses:
The authors present an interesting and well written manuscript on the synthesis to produce GO using a low-cost and environmentally friendly modified hummers method. The rationale for conducting this work is well-justified and the paper aligns nicely with the scopes of the journal. The paper is well-organized and flows smoothly. I would recommend the following points before publication:
- In the Raman analysis, please include the ID/IG ratio that shows the functionalization degree. Also from the Raman analysis you can calculate the number of flakes.
The intensity of radio ID/IG and the number of layers in the flakes were included. Adding the necessary references
- The thickness of the GO nanosheet is of significance importance. Please include an AFM measurement and statistical analysis of thickness and lateral dimension. This is very important due to the novelty of the synthesis; the readers must have all the information about the quality/morphology of the nanosheets. From the SEM you cannot have an estimation of the thickness. In your images you have a network not an isolated nanosheet.
Thanks for the suggestion, however it is not possible to carry out AFM measurements since my institution does not have the necessary equipment. The text was corrected indicating that the thickness estimate is for the layer network in the flakes.
- You should write a paragraph in the introduction about the novelty of your work. It’s not very wise for your good work to have only 2-3 sentences.
The novelty of the work was added in the introduction
- The following references, they will help you enhancing the quality of your MS. Please include them.
The following references were included
- Graphene Oxide–Cytochrome c Multilayered Structures for Biocatalytic Applications: Decrypting the Role of Surfactant in Langmuir–Schaefer Layer Deposition (doi: 10.1021/acsami.2c03944)
- Controlled deposition of fullerene derivatives within a graphene template by means of a modified Langmuir-Schaefer method (doi: 10.1016/j.jcis.2018.04.049)
- Mapping of graphene oxide and single layer graphene flakes—defects annealing and healing (doi: 10.3389/fmats.2018.00037)

Reviewer 3 Report
This work may be considered for publication after addressing the following comments carefully. I would like to see the revised version before its acceptance.
1. The comparison of the present approach with the traditional method should be provided as a Table.
2. Authors should discuss the possible impurities in the prepared solids. These should also be compared with the previous works.
3. Elemental analysis is missing and it should be reported.
4. TEM measurement should be performed and to be discussed.
5. Figure qualities should be drastically improved with better legends, reducing the noise in IR and Raman spectra. Also avoid typographical error (Wave Number!).
6. The manuscript should be extensively revised to eliminate typographical errors and English should be improved.
Author Response
Editor-in-Chief
Materials
We sincerely appreciate that you have taken the time to review our work Eco-friendly approach for graphene oxide synthesis by modified Hummer’s method. We welcome your comments which are all valuable and helpful in improving the quality of our article.
The observations have been carefully studied, and, in the same way, the proper corrections have been made, hoping that they will meet your expectations and thus have your approval. The following pages of this document respond to the indications of each reviewer.
The text highlighted in green corresponds to each of the observations made by the reviewers, while the text highlighted in yellow shows the correction made by the authors. The authors welcome any further observations or future comments.
Dr. Néstor Méndez Lozano
(corresponding autor)
Reviewer 3
We appreciate that you have shared your valuable time and knowledge to review our work. Your respected recommendations have been studied extensively and the responses we have made are shown below.
This work may be considered for publication after addressing the following comments carefully. I would like to see the revised version before its acceptance.
- The comparison of the present approach with the traditional method should be provided as a Table.
The table was added comparing the traditional method vs modified Hummer's method
- Authors should discuss the possible impurities in the prepared solids. These should also be compared with the previous works.
The presence of impurities such as potassium and sulfur were discussed. Other studies also report the presence of impurities due to the synthesis process.
- Elemental analysis is missing and it should be reported.
EDS analyzes were included to determine the elemental composition
- TEM measurement should be performed and to be discussed.
TEM analyzes were not possible. In my institution we do not have a transmission microscope and its use in other laboratories is saturated.
- Figure qualities should be drastically improved with better legends, reducing the noise in IR and Raman spectra. Also avoid typographical error (Wave Number!).
The quality of the figures was improved as well as the typographical errors
- The manuscript should be extensively revised to eliminate typographical errors and English should be improved.
English style corrections were added

Round 2
Reviewer 3 Report
It can be considered for publication.